# Readout of a solid state spin ensemble at the projection noise limit

Rouven Maier [1,2], Cheng-I Ho[1,3], Andrej Denisenko[1], Marina Davydova[4], Peter Knittel [4], Jörg Wrachtrup [1,2,3] & Vadim Vorobyov [1,3] ✉

Spin ensembles are central to quantum science, from fundamental physics searches to magnetic resonance spectroscopy and quantum sensing. The standard quantum limit for their performance is ultimately posed by spin projection noise, yet solid-state implementations have so far been limited by significantly larger photon shot noise. Here, we demonstrate a direct quantum non-demolition readout of a mesoscopic ensemble of nitrogen-vacancy (NV) centers in diamond that surpasses the photon shot-noise limit and approaches the intrinsic spin projection noise. By stabilizing the intrinsic $^{14}$N nuclear spin bath at high magnetic fields and employing a repetitive nuclear-assisted spin readout, we achieve a noise reduction of 3.8 dB below the thermal projection noise level. This enables direct access to the intrinsic fluctuations of the spin ensemble, allowing us to directly observe the signatures of correlated spin states. Our results establish projection noise-limited readout as a practical tool for solid-state quantum sensors, opening pathways to quantum-enhanced metrology, direct detection of many-body correlations, and the implementation of spin squeezing in mesoscopic solid-state ensembles.

Ensembles of quantum spins present a foundational platform in quantum physics with a wide range of applications, ranging from inertial navigation[1], searches for dark matter[2], to widespread nuclear magnetic resonance (NMR) spectroscopy[3,4] and magnetic resonance imaging (MRI)[5,6] applications. Atomic spin ensembles[7–9] are the gold standard with regard to magnetic field sensitivity due to sub-projection noise-limited readout[10–13] and long coherence times, allowing for example, the detection of human brain activity in vivo[14]. Following an early theoretical proposal[15], this efficient readout in combination with a high degree of spin control paved the way to even overcome the projection noise limit (standard quantum limit) by applying spin squeezing[16–18]. Extending these capabilities to solid-state spin ensembles enables scalability and integrable quantum sensors[19–21] with applications ranging from nanoscale magnetic resonance spectroscopy, bioimaging[22–26], and nanoscale scanning probe[27,28] to studies of correlated sensors[29–31]. Among solid-state platforms, nitrogen-vacancy (NV) centers in diamond combine long coherence times,

room-temperature operation, and optical addressability[32]. Yet, unlike their atomic counterparts and despite significant progress in solid-state ensembles, including NVs, readout schemes have so far remained constrained by photon shot noise[33–36], obscuring intrinsic quantum fluctuations. This restriction not only limits the sensor performance, but also precludes direct observation of collective quantum effects such as correlated spin noise or spin squeezing[37–39].

Here, we overcome this barrier by realizing a projection noise-limited readout in a solid-state spin ensemble. Utilizing repetitive nuclear-assisted measurements, formerly developed for single spins[40,41], and applying it to a mesoscopic spin ensemble of NV centers at high magnetic fields, we achieve a noise reduction of 3.8 dB below the photon shot noise level. We probe the crossover between photon shot noise and spin projection noise-limited regimes in the optical readout of the unpolarized ensemble and show the traces of correlated states in the variance of the readout noise distribution and propose it for correlated quantum sensing using sub-micron-scale ensembles of

[1]3rd Institute of Physics, University of Stuttgart, Stuttgart, Germany. [2]Max Planck Institute for Solid State Research, Stuttgart, Germany. [3]Center for Integrated Quantum Science and Technology (IQST), Stuttgart, Germany. [4]Fraunhofer Institute for Applied Solid State Physics (IAF), Freiburg, Germany. ✉e-mail: v.vorobyov@pi3.uni-stuttgart.de

NV centers. The sensitivity improvement up to several orders of magnitude for quantum sensing protocols through projection noise-limited readout is discussed and the optimum readout conditions are identified. By closing the gap between atomic and solid-state ensemble measurements, our work establishes projection noise-limited readout as a new resource for solid-state quantum sensing and lays the foundation for quantum-enhanced metrology, correlated sensing, and many-body studies in diamond spin ensembles.

## Results

In our experiments, an ensemble of nitrogen vacancy centers is optically read out under ambient conditions (Fig. 1). The average collective nuclear spin $\tilde{J}_z = \frac{J_z}{N_{NV}} = \frac{1}{N_{NV}} \sum_{i=1}^{N_{NV}} I_z^{(i)}$ of the $N_{NV}$ intrinsic nitrogen spins with spin $I = 1$ of an ensemble of NV centers is measured by optical quantum-non-demolition measurements[40], where $I_z^{(i)}$ is the spin projection along the $z$ axis of the individual nitrogen spin $i$. Spin state readout of the nitrogen hyperfine levels is achieved by repeated mapping onto the reinitialized electron spin eigenstates via a spin-selective narrowband MW $\pi$ pulse (Supplememtary information URL provided by the journal later), which is then read out and reinitialized repeatedly by applying a laser pulse (Fig. 2a, b). Subtracting the photon numbers $a$ and $b$ associated with the population of the different spin states, yields a Skellam distribution, with photon shot noise limited standard deviation (Fig. 2c)

$$\sigma_n = \sqrt{2n\left(1 - \frac{c}{2}\right)}, \tag{1}$$

where $c$ is the optical contrast of the readout and $n$ is the average number of photons detected (see "Methods").

However, the detected photon distribution is in general, a convolution of the pure photon statistics and the underlying spin distribution. Thus, the observed standard deviation is a variance sum of the photon shot noise $\sigma_n$ and the spin projection noise $\sigma_{\tilde{J}_z}$:

$$\sigma^2 = \sigma_n^2 + (2nc\sigma_{\tilde{J}_z})^2, \tag{2}$$

demonstrating that for large photon numbers the spin projection noise dominates.

In the following, we analyze the standard deviation of the obtained signal multiplied by the scaling factor $\sigma' = \sigma/2nc$ (see "Methods"). By gradually increasing the number of detected photons $n$ through an increase of the number of repetitive readouts $m$ from 1250 to 25,000, we probe the crossover in the readout of the spin ensemble from photon shot noise to spin projected noise-limited regime depicted on Fig. 2d. In the limit of small $n$, the normalized standard deviation is dominated by the photon shot noise and scales as $\sigma' \to 1/\sqrt{n}$. By increasing the number of readout repetitions, $\sigma'$ begins to deviate from the photon shot noise scaling and levels off when reaching the thermal spin projection noise given by $\sigma' \to \sqrt{\frac{I(I+1)}{3N_{NV}}}$ (see Supplementary Information). In this regime, the direct readout of the ensemble spin state statistics becomes possible. In our experiment, the readout noise surpasses the spin projection noise by 3.8 dB. The maximum number of readout repetitions is limited by the effective longitudinal relaxation time $T_1$ of the probed spins. As it increases quadratically with the applied magnetic field[40], we apply a magnetic field of $B_0 = 2.7$ T, which enables the realization of more than the ~4000 repetitive readouts required to reach the projection noise limit. To account for residual spin relaxation during the readout (Fig. 2d), we adapt a model which captures the reduction in spin projection noise under slightly perturbing measurements (see "Methods"). Overall, the projection noise $\sigma_{\tilde{J}_z}$ in the measured data can be fitted as a function of the number of nitrogen spins $N_{NV}$ as:

$$\sigma_{\tilde{J}_z} = \sqrt{\frac{1}{3}}\sqrt{\frac{I(I+1)}{3N_{NV}}}\sqrt{\frac{2T_1^2}{T^2}\left(\frac{T}{T_1} + e^{-T/T_1} - 1\right)}. \tag{3}$$

As a result, this method enables the direct determination of active NV centers in the confocal spot of the laser from the model fit parameters ($N_{NV} = 31(3)$). We probe it in three various NV density spots on our sample, yielding a clear linear correlation with observed fluorescence levels (see Supplementary Information). It also should be noted that as the relative photon shot noise and the thermal projection noise both decrease as $\sim 1/\sqrt{N_{NV}}$, the readout crossover is independent of the

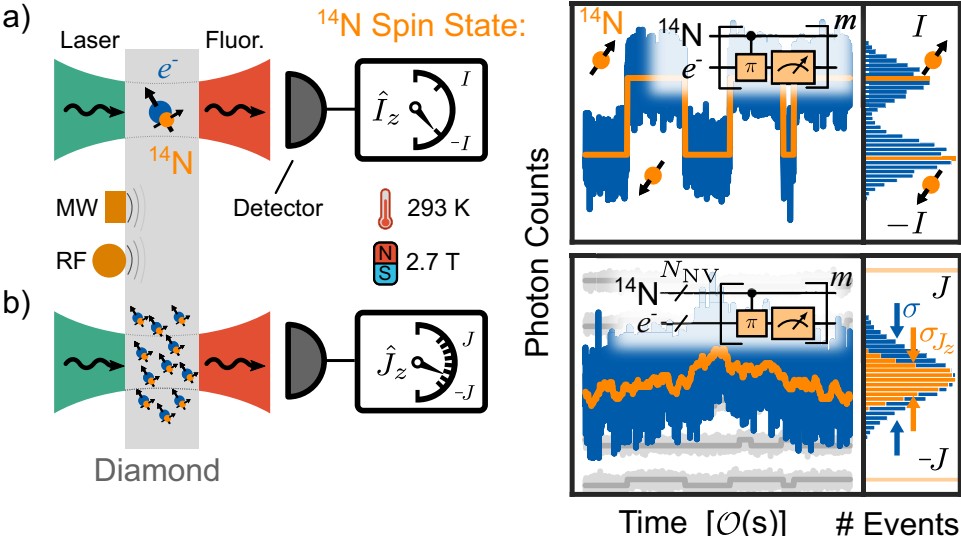

**Fig. 1 | Modeled quantum state readout: From single spins to spin ensembles. a** A single Nitrogen-vacancy (NV) center in a confocal microscope at a magnetic field of $B_0 = 2.7$ T is controlled by laser, microwave (MW) and radio frequency (RF) pulses and the fluorescence is read out by the detector. The Nitrogen spin $I_z$ undergoes stochastic spin fluctuations, producing a random telegraph signal (orange line). Photon shot noise (blue line) is introduced by the optical readout. **b** In the signal of an ensemble of NV centers, the individual telegraph like signals (gray lines) are summed to a net magnetization $\langle J_z \rangle$ (orange line), where individual spin states can no longer be determined. Typically, the fluctuations $\sigma_{\tilde{J}_z}$ (projection noise) are hidden in the photon shot noise (blue line).

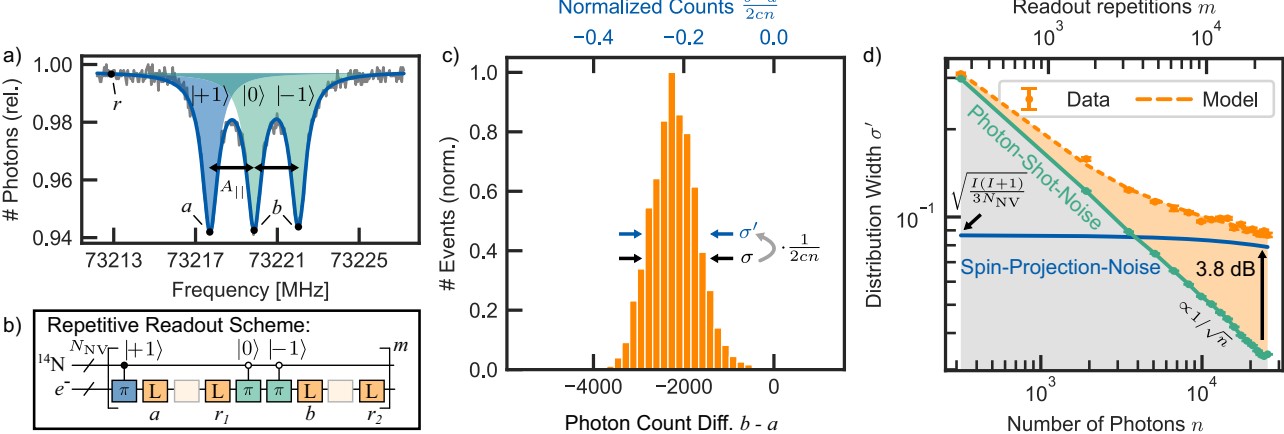

**Fig. 2 | Surpassing the photon shot noise. a** Optically detected magnetic resonance (ODMR) spectrum of the NV center. The nuclear spin-selective electron transitions are separated by the hyperfine coupling $A_{\parallel} = 2.16$ MHz. **b** Pulse sequence of the repetitive readout of $N_{NV}$ nuclear spins. The nuclear spin states are transferred onto the electron spin via selective $\pi$-rotations, before a readout laser pulse is applied. The detected photons ($b - a$) are summed up over $m$ repetitions of the readout. Reference measurements ($r$) are added in between the spin readouts. The respective photon counts $a$, $b$ and $r$ are visualized in (**a**). **c)** Typical photon histogram of 3000 experiments obtained by $b - a$. The (normalized) width ($\sigma'$) $\sigma$ is

caused by the photon shot noise and the spin projection noise. **d** An increase of photon counts $n$ during the readout reduces the photon shot noise (green) as $1/\sqrt{n}$. For high photon counts, a transition from the photon shot noise-dominated regime to the spin projection noise-dominated regime (blue) is observed, where the photon shot noise is 3.76(1) dB lower than the observed projection noise. The projection noise is given by the statistical polarization $\sigma_{J_z} = \sqrt{\frac{I(I+1)}{3N_{NV}}}$, which reduces during the readout due to effective $T_1$ relaxation of the nitrogen spins. Error bars represent the standard error of the mean.

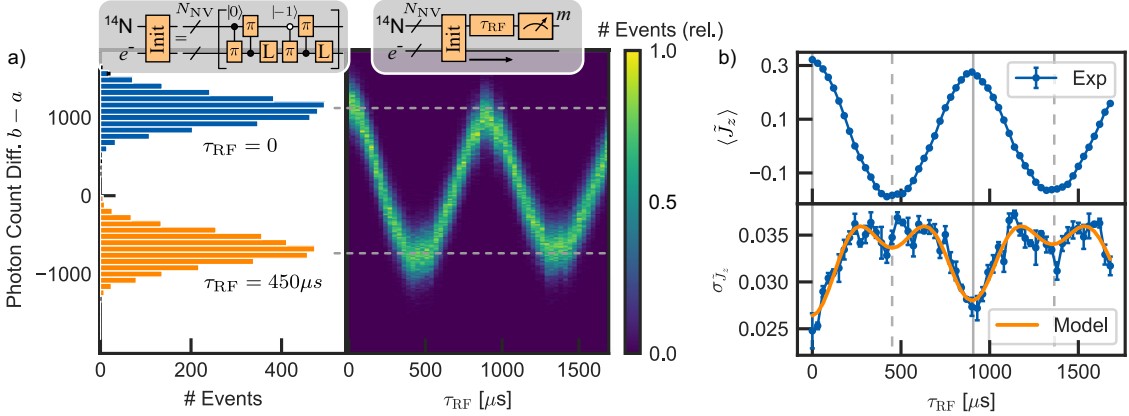

**Fig. 3 | Nitrogen spin control. a** The ensemble of nitrogen spin states can be polarized by swapping the polarized electron spin state onto the nitrogen spins with a series of controlled $\pi$ rotations. Rabi oscillations of the nitrogen spins are driven by resonant RF. The histograms of the detected photons $b - a$ shift accordingly. Full inversion is not achieved due to a limited initialization fidelity of the electron spin. Error bars represent the square root of the number of photons.

**b** The extracted spin distribution width $\sigma'$ follows the oscillations of the spin projection noise $\sigma_{J_z}^2 \propto p(1 - p)$, where $p$ is the readout probability of the nitrogen eigenstates. Full recovery of the polarized $\sigma_{J_z}$ after a full inversion (dashed line) is not observed due to faster spin decay in the nitrogen $|0\rangle$ state during the readout (see main text). Error bars represent the standard error of the mean.

size of the spin ensemble (Supplememtary information URL provided by the journal later.).

With the established readout, we investigate the coherent control of the nuclear spin ensembles at hand. After initialization into $|+1\rangle$ (see "Methods"), applying resonant radio frequency (RF) pulses allows coherent spin control of the nitrogen spin, as long-lived Rabi oscillations are observed (Fig. 3a). A deviation of the average spin state from $\langle \tilde{J}_z \rangle = 0.5$ after the initialization is due to an initialization infidelity of the nitrogen spin states, with some remaining population in $|0\rangle$ and $|-1\rangle$. Additionally, limited initialization fidelity of the electron spin prevents a full population inversion, as the RF is only resonant to the nitrogen transition if the electron is initialized into the correct charge and spin state (i.e., NV$^-$; $m_s = 0$). The other fraction of nitrogen spins stays polarized during the experiment. Due to repetitive initialization

of the electron spin during the readout step, we still read out the spin state of all nitrogen spins. As a result of the projection noise-limited readout, we are able to track the spin projection noise $\sigma_{J_z}$ in addition to the average spin state $\langle \tilde{J}_z \rangle$ during the drive of the nuclear spin oscillations (Fig. 3b). The projection noise $\sigma_{J_z}$ is directly calculated according to Eq. (5). The spin projection noise follows the expected projection noise relation of $\sigma_{J_z} \sim \sqrt{p(1-p)N_{NV}}$, where $p$ is the probability to measure the eigenstates. Starting with a fully polarized state, the width of the distribution is minimized, limited by a residual decay of the spin state during the readout and the infidelity of the polarization. During the Rabi oscillations, the width reaches its maximum when the ensemble state crosses the equator plane, before it is reduced again, following the partial inversion of the population. The limited dip

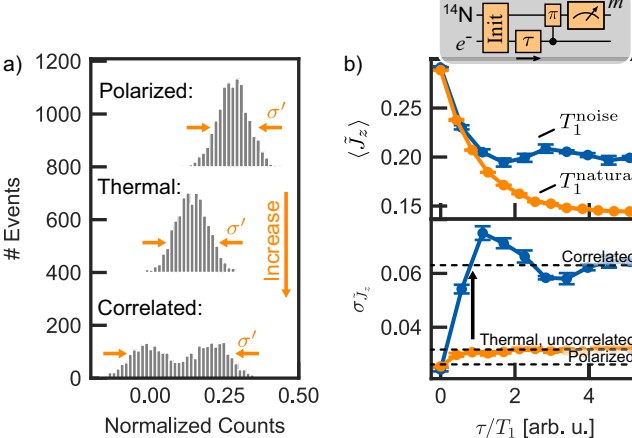

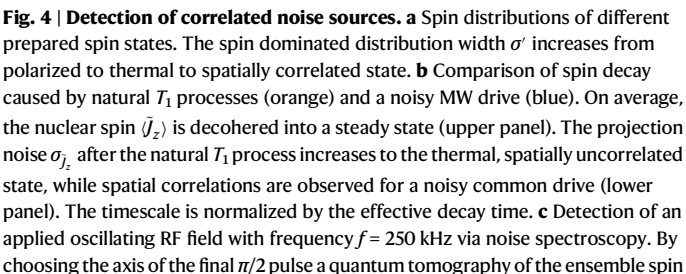

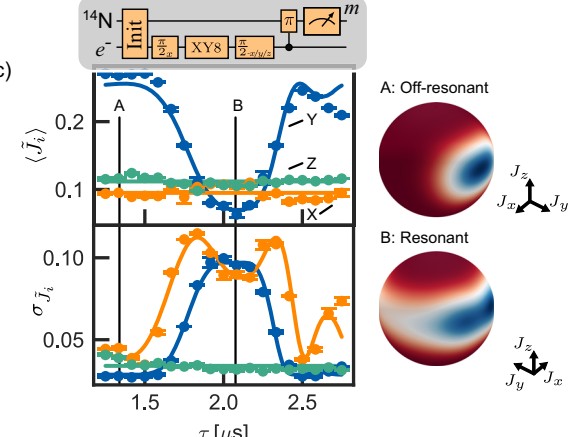

**Fig. 4 | Detection of correlated noise sources. a** Spin distributions of different prepared spin states. The spin dominated distribution width $\sigma'$ increases from polarized to thermal to spatially correlated state. **b** Comparison of spin decay caused by natural $T_1$ processes (orange) and a noisy MW drive (blue). On average, the nuclear spin $\bar{J}_z\rangle$ is decohered into a steady state (upper panel). The projection noise $\sigma_{\bar{J}_z}$ after the natural $T_1$ process increases to the thermal, spatially uncorrelated state, while spatial correlations are observed for a noisy common drive (lower panel). The timescale is normalized by the effective decay time. **c** Detection of an applied oscillating RF field with frequency $f$ = 250 kHz via noise spectroscopy. By choosing the axis of the final $\pi/2$ pulse a quantum tomography of the ensemble spin

state along X (orange), Y (blue) and Z (green) after interaction with the external field is achieved. When coupling to the external field (at point B), the polarization along Y is destroyed, while X and Z remain unpolarized (upper panel). The correlated spin distribution is observable and shows an increase in $\sigma_{j_x}$ and $\sigma_{j_y}$ as the spins are redistributed in the equatorial plane through the interaction with the oscillating signal (lower panel). The Husimi $Q$-function of the reconstructed spin distribution is visualized on a sphere, showing the delocalization of the spin states in the equatorial plane. For details about the fitting model (solid lines) and the spin state reconstruction, see Supplementary Information. Error bars represent the standard error of the mean.

of $\sigma_{\bar{J}_z}$ at the inversion is due to an increased polarization decay $T_1$ of $|0\rangle$ compared to $|+1\rangle$ during the readout (see Supplementary Information).

Next, we employ the readout to explore novel sensing capabilities of the electron spin ensemble for spatially correlated sensing. By partially swapping the electron spin state after a sensing protocol to the initialized nitrogen spin state, nuclear spin-assisted electron spin readout can be performed and is applied to two typical sensing protocols: $T_1$ relaxometry[42,43] and noise spectroscopy[44,45] (Fig. 4).

In the case of $T_1$ relaxometry, we compare spatially correlated noise created by a wideband microwave signal and uncorrelated intrinsic phonon-induced noise and investigate their influences on the ensemble spin state. For a correlated noise source, a microwave signal generated as a sum of ten fully randomized frequency sources in a 3 MHz band around the NV center electron spin resonance transition frequency is applied. This induces fast electron spin decay. For the uncorrelated noise source, we rely on the intrinsic phonon-induced translational relaxation of electron spin sublevels, as they occur independently and incoherently for each spin in the ensemble. The observed spin decay and its standard deviation are plotted in Fig. 4a, b. While the average spin state $\langle\bar{J}_z\rangle$ decays similarly, a distinction in the obtained $\sigma_{\bar{J}_z}$ is seen, indicating a clear sign of correlations induced by the correlated noise source. The decay curve of the noise-induced experiment follows the expected average of a randomly driven spin. The thermal spin distribution induced by uncorrelated noise is significantly narrower compared to the spin distribution induced by the correlated noise. This shows that having access to the direct spin readout can allow the investigation of spatial correlations in environmental effects using sub-micron-scale ensembles of sensors in microscopy applications.

In a second typical scenario, $T_2$ relaxometry is used to detect correlated oscillating noise generated by a monochromatic source with a stochastic phase $S = B_{osc}\cos(2\pi f t + \lambda)$, with frequency $f$ = 250 kHz and amplitude $B_{osc}$ = 1.84(7) µT at the position of the NV center. After preparing the initial electron spin state $\rho_S = (\mathbf{1}_e + \sigma_y/2)^{\otimes N}$ a XY8

dynamical decoupling sequence is applied and the interaction with the noise is controlled by adjusting the inter-pulse spacing $\tau$ with resonance at $\tau = 1/(2f)$[46]. The marginal spin projection distributions $\tilde{J}_x, \tilde{J}_y, \tilde{J}_z$ after the interaction with the noise are obtained by applying different pulses $\{\pi/2_y, \pi/2_{-x}, \varnothing\}$ followed by $\langle\tilde{J}_z\rangle$ readout, as shown in Fig. 4c. Apart from the usually detected average value $\langle\tilde{J}_i\rangle$ the correlated spin fluctuations along the X, Y, and Z axis induced by the correlated oscillating noise are probed. Again, direct access to the spin distributions by projection noise-limited readout allows for a more sophisticated study of the effects of the noisy environment on the sensor spins, by clearly showing delocalization in the equatorial plane. To better visualize the ensemble spin state, we plot the Husimi $Q$-function of the regularized reconstructed spin distribution of the active spins on a sphere. For details regarding the reconstruction method, see Supplementary Information.

After establishing a readout technique that can reach the spin projection noise level, we investigate its significance for common quantum sensing protocols with sensing time $\tau_{sens}$. The relevant figure of merit is the sensitivity $\eta$, which is given by:

$$\eta \propto \frac{1}{c_{eff} n \sqrt{\tau_{sens}}} \sqrt{\frac{\tau_{sens} + \tau_{other}}{\tau_{sens}}} \sigma, \tag{4}$$

where the effective contrast $c_{eff}$ and experimental overhead $\tau_{other}$ (e.g., initialization and readout) are counteracting the positive effect of a reduced noise level $\sigma$. Based on the experimental parameters extracted in this work (see Supplementary Information), the sensitivity of sensing protocols with a conventional readout consisting of a single direct electron spin readout with a laser pulse is compared to the sensitivity of the nuclear spin-assisted repetitive readout. For example, a sensing protocol in the linear regime around the sensing state $\rho_S = (\mathbf{1}_e + \sigma_y/2)^{\otimes N}$ (i.e., maximum spin projection noise) is investigated. The ratio $\eta_{conv}/\eta_{rep}$ as a function of the number readouts $m$ and the duration of the sensing protocol $\tau_{sens}$ is shown in Fig. 5a. Already for sensing protocols with $\tau_{sens} > 15$ µs, which is typical for $T_2$ based sensing protocols, the

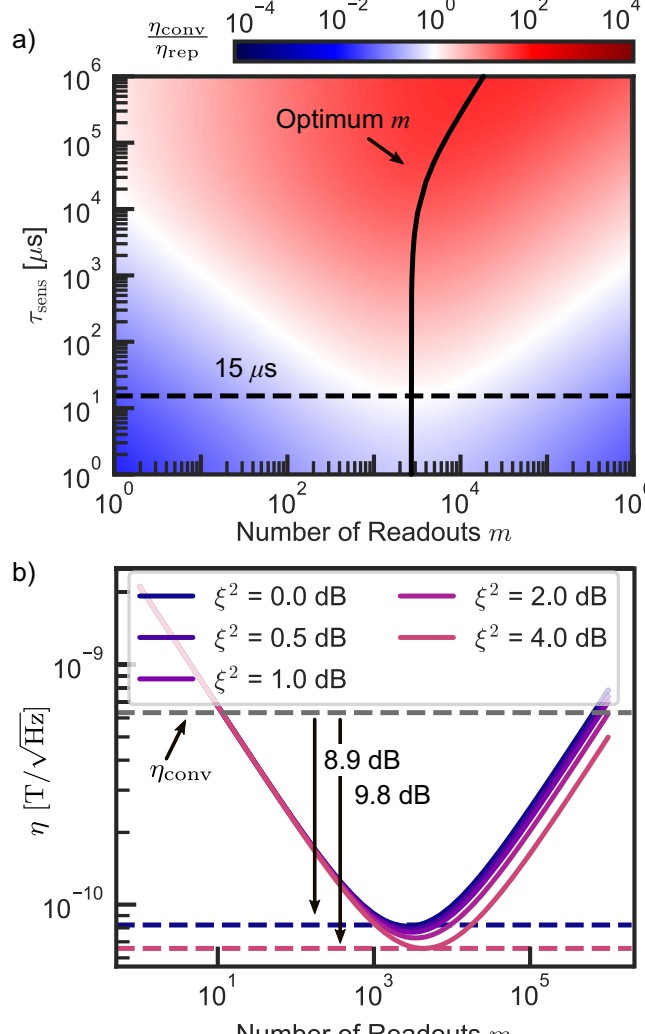

**Fig. 5 | Calculated sensitivity improvement for quantum sensing applications through projection noise limited readout. a** For sensing protocols with $\tau_{\text{sens}} > 15\,\mu s$ the repetitive readout scheme improves sensitivity of a superposition state readout (red area) compared to the conventional, single readout. The optimum sensitvity is reached at $m = 2700$ readouts for $\tau_{\text{sens}} < 10000\,\mu s$. For larger $m$, the spin projection noise limit is reached, limiting signal improvement, while the protocol duration is increased, leading to a declining sensitivity. **b** Spin squeezing $\xi^2$ can further improve the sensitivity by reducing the spin projection noise limit. Here, the absolute values for $N_{\text{NV}} = 100$ and $\tau_{\text{sens}} = 1\,\text{ms}$ are shown.

repetitive readout can improve the sensitivity, while for very long measurement sequences the improvement can reach several orders of magnitude. Already for magnetic fields larger than 0.5 T, the optimum number of readouts of $m \sim 2700$ can be reached within the effective $T_1$ of the nitrogen memory, assuming a quadratic field dependence (see Supplementary Information URL). For a given $\tau_{\text{sens}}$, $\eta_{\text{rep}}$ first improves as $\sigma$ approaches the spin projection noise limit. After this point, $\sigma$ cannot be improved further by increasing $m$, while simultaneously increasing the experimental overhead, leading to an increase of $\eta_{\text{rep}}$. One avenue to further increase the sensitivity of spin projection noise-limited readouts is the use of spin-squeezing[37]. The influence of spin squeezing up to 4 dB on $\eta_{\text{rep}}$ as a function of the readout repetitions is shown in Fig. 5b. However, previously demonstrated experimental values of $\eta^2 = 0.5$ dB in solid state ensembles[37] only lead to a minor sensitivity improvement of 0.13 dB.

## Discussion

In summary, we have realized projection noise-limited readout of a solid-state spin ensemble, surpassing the thermal spin projection noise floor by 3.8 dB. This advance provides a long-sought capability: direct access to the fundamental fluctuations of solid-state mesoscopic spin systems.

Our approach can improve the sensitivity of NV-based quantum sensors up to several orders of magnitude and opens the opportunity to harness the benefits of spin-squeezing in solid-state quantum sensing. This will enable rapid NMR, MRI, magnetometry and relaxometry measurements based on quantum sensors, where long averaging times are limiting the applicability. Another exciting avenue is the microscopy of temporally and spatially correlated signals, enabled by projection noise-limited readout. We envision a widefield microscope, where each diffraction-limited pixel is detecting spin distributions, which can directly be analyzed to obtain a correlation map. Possible areas of application include the investigation of interaction types in magnetic materials, phase transitions or electronic chip failures. Further, the application of solid-state spin ensembles in the field of quantum simulations[47] is directly linked to the readout of the spin states. A projection noise-limited readout, for example, allows the extraction of 2nd, 4th, and higher order cumulants to observe non-trivial, non-classical, and non-Gaussian many-body spin states[48].

## Methods

All experiments are carried out on a home-built confocal microscope at room temperature (Fig. 1). The (111) cut diamond chip containing as grown ($\delta$-doped) preferentially aligned thin ensembles of $^{14}$NV centers is placed in a superconducting magnet, with a bias magnetic field of $B_0 = 2.7$ T aligned to the principal NV axis. Microwave (MW) control of the electron spin at 73 GHz is achieved using a $\lambda/2$ bow-tie resonator, as described in ref. 49. Nuclear spins are controlled via radio frequency (RF) from a 50 μm copper wire spanned across the sample. Emitted photons are detected using an avalanche photodiode (APD). See Supplementary Information for further details.

### Data analysis

The optical readout of the nitrogen spin states $|+1\rangle$, $|0\rangle$ and $|-1\rangle$ is performed via mapping onto the two electron spin states $|0\rangle_e$ and $|-1\rangle_e$. As $|0\rangle$ and $|-1\rangle$ are mapped onto the same electron spin states, the three nitrogen states are mapped onto a pseudo spin-1/2 system, with the associated readout counts $a$ and $b$. Additionally, the optical readout of the electron spin states are interspersed with two reference readouts of the baseline photon counts $r_1$ and $r_2$, to obtain the mean baseline photon counts $n = \frac{r_1+r_2}{2}$. The total optical readout contrast for each measurement is calculated via $c = \left(\frac{n-a}{n}\right) + \left(\frac{n-b}{n}\right)$. To increase the signal contrast and remove the influence of slow drifts of laser power, the main signal is obtained by subtracting the counts $b - a$. The nuclear spin state is left almost undisturbed by this process, allowing many repetitions $m$ of the readout to accumulate fluorescence signal of the photon count difference $b - a$. To normalize the readout to the maximum possible spin readout $[I, -I]$, it is normalized by $2nc$ throughout the analysis: $a - b \to \frac{a-b}{2nc}$ and therefore $\sigma \to \sigma' = \frac{\sigma}{2nc}$.

This photon distribution enables the reconstruction of the nuclear spin distribution properties as described below. The average $\langle \tilde{J}_z \rangle$ is given simply by a linear expression $\langle \tilde{J}_z \rangle = k_0 \langle b - a \rangle$, where $k_0 = 1/2nc$, with $c$ being the optical contrast of the readout between states $|0\rangle^{\otimes N_{\text{NV}}}$ and $|+1\rangle^{\otimes N_{\text{NV}}}$. The influence of the spin projection noise $\sigma_{\tilde{J}_z}$ and photon shot noise $\sigma_n$ results in a variance sum for the observed signal distribution:

$$\sigma^2 = \sigma_n^2 + (2nc\sigma_{\tilde{J}_z})^2. \tag{5}$$

The variance of the readout distribution is given by the variance sum of the spin projection noise $\sigma_{\tilde{J}_z}$ and the photon shot noise $\sigma'_n$. Due to the high number of detected photons, the APD is no longer operated in the linear regime, resulting in an artificial decrease in the width of the detected photon distribution. This is accounted for by adding a brightness-dependent verified correction factor $k$ to the model ($k \sim 0.89$–$0.99$, see Supplementary Information), yielding:

$$\sigma' = k\sqrt{\left(\sigma'_n\right)^2 + \sigma_{\tilde{J}_z}^2}. \tag{6}$$

The spin projection noise of a thermal spin ensemble is given by

$$\sigma_{\tilde{J}_z} = \sqrt{\frac{I(I+1)}{3N_{\mathrm{NV}}}}. \tag{7}$$

However, residual spin state decay during the extended readout period has to be taken into account for a quantitative variance analysis. First, we estimate the correlation function of the spin ensemble under measurements $C(\tau) = \langle \tilde{J}_z(t)\tilde{J}_z(t+\tau)\rangle_t = \sigma_0^2 e^{-|\tau|/T_1}$ as an exponential decay with characteristic time $T_1$, confirmed in a separate measurement (see Supplementary Information). Second, the standard deviation of the spin distribution under perturbing measurement is estimated as a standard deviation of a stochastic process averaged over measurement time $T \sim T_1$ comparable to the relaxation time (see[50] and Supplementary Information for more details). The projection noise $\sigma_{\tilde{J}_z}$ in the measured data can be fitted as a function of the nuclear spin relaxation time $T_1$ under readout and the number of nitrogen spins $N_{\mathrm{NV}}$ as:

$$\sigma_{\tilde{J}_z} = \sqrt{\frac{1}{3}}\sqrt{\frac{I(I+1)}{3N_{\mathrm{NV}}}}\sqrt{\frac{2T_1^2}{T^2}\left(\frac{T}{T_1} + e^{-T/T_1} - 1\right)} \tag{8}$$

The factor $\sqrt{1/3}$ is introduced as a side effect of reading out the population of the three nitrogen spin states by mapping them onto only two electron spin states (see Supplementary Information).

## Data availability

The source data related to Figs. 2–5 are publicly available at DaRus[51].

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

## Acknowledgements

We acknowledge funding from the German Federal Ministry of Education and Research (BMBF) through the projects Clusters4Future QSens and DiaQnos, and the German Federal Ministry of Research, Technology and Space through projects Qsolid and QSi2V. We further acknowledge funding from the European Union through project C-QuENS (grant agreement no. 101135359), Amadeus (grant agreement no. 101080136), and Spinus (grant agreement no. 101135699), as well as the Carl-Zeiss-Stiftung via QPhoton Innovation Projects and the Center for Integrated Quantum Science and Technology (IQST). R.M. acknowledges support from the International Max Planck Research School (IMPRS).

## Author contributions

R.M. and V.V. conceived the idea, planned the experiment, discussed the data, and prepared the manuscript with input from all co-authors. R.M. and C.H. performed the main measurements. M.D. and P.K. manufactured the diamond sample. C.H. and A.D. performed preliminary measurements. J.W. and V.V. led the overall project.

## Funding

## Competing interests

The authors declare no competing interests.
