## [Transparent Peer Review file · Nature Communications]

Readout of a solid state spin ensemble at the projection noise limit

Corresponding Author: Dr Vadim Vorobyov

Version 0:

Reviewer comments:

Reviewer #1

(Remarks to the Author)

The authors present work improving the sensitivity of readout of an ensemble of nitrogen vacancy nuclear spins, leveraging techniques employed for the readout of single spins. Operating at a higher than typical magnetic field, the photon shot noise is measured to be 3.8 dB lower than the spin-projection noise following repeated readout. This is significant as it opens up the possibility of observing spin squeezing in a room-temperature ensemble as well as measuring noise correlations across the ensemble. The latter is exemplified through a simple T2 relaxometry experiment examining the spin projection under a phase-randomised stimulus.

Overall, the work is reasonably presented and to my knowledge is the first demonstration of measurements of a room temperature spin ensemble at the quantum limit in the solid state. The choice of data collection and analysis methods are sound. I believe this manuscript is appropriate for this journal and will be useful to the community, though some minor revision may be needed.

Below are some technical questions and comments I have.

For the ODMR detection of the nuclear spin states, a series of electron rotations were performed at ~73 GHz. It wasn't mentioned what the Rabi frequency of the electron spin was, but the frequency upconversion and amplification chain, along with the pulse modulation can lead to spurious excitation of the neighbouring ESR lines (especially for a narrow separation of $A_{\text{parallel}} = 2.16$ MHz). Can the authors quantify this readout mapping error?

On the topic of the readout, the perpendicular component of the hyperfine interaction was not listed, but the "T1" decay measurements are in fact dominated by hyperfine-enhanced relaxation, as opposed to direct spin-lattice relaxation. Were any measurements performed to measure the hyperfine strength transverse to the external field? This has significant implications in the discussion of the increasing number of readout shots performed, because this relaxation term scales with the number of shots m , rather than with time.

Finally, the T1 decay is shown to reduce the spin-projection noise due to depolarisation. Conceivably performing a negative result measurement, i.e. monitoring the expected dark-state, would reduce the impact of readout allowing the spin-photon level to be reduced further while maintaining the spin-polarisation. There is recent work on this performed on donor spins in silicon (see below).

Refs:

Avshalom C Elitzur and Lev Vaidman. Quantum mechanical interaction-free measurements. *Foundations of physics*, 23(7):987–997, 1993.

Maximizing the nondemolition nature of a quantum measurement via an adaptive readout protocol
<https://arxiv.org/abs/2511.10978>

Detailed comments:

- Fig 2d, the green photon shot-noise curve should not have datapoints as it is not data.
- The introduction of c , r_1 and r_2 is unclear. I believe r_1 and r_2 are the reference measurements bookending each sequence, but this needs to be explicit. The given form for c is not intuitive.
- Fig 3a, where are the two histograms derived from? It appears they are from different τ_{RF} slices, but this is not

mentioned.

Reviewer #2

(Remarks to the Author)

Several works in atomic physics have demonstrated the capability to beat the measurement shot noise and go beyond the standard quantum limit or the projection noise limit. However, solid-state systems, where readouts rely on fluorescence differences due to spin-dependent intersystem crossing rates, have been plagued by “low information” per photon count and thus more susceptible to photon shot noise. This work demonstrated one path towards reaching the spin projection noise limit by exploiting long-coherence-time nitrogen-14 nuclear spins and their hyperfine interactions with the electronic spins.

The authors worked with an NV center ensemble subjected to high magnetic field (~2.7 T). Nuclear-assisted readout protocol was used to measure the polarization of nitrogen-14 by mapping the N-14 spin population to the electron. While this is a conventional protocol for exploiting the weakly interacting N-14 as the long-lived “memory”, the authors explored the large field regime where they can afford a large number of readout cycles to average down the photon shot noise and reveals the cross-over from shot-noise-limited to spin-projection-noise-limited statistics. Having this tool at hand, the authors demonstrated a measurement of spatially correlated spin ensembles and an enhanced quantum tomography.

Overall, the paper is well-written and the derivations are sound. I would like to recommend the paper for publication in Nature Communications.

There are a few small questions that may improve the paper further:

1. Nuclear T_1 is extended significantly due to high magnetic field for repeated readouts. Would it be possible to show in Fig. 5 a) the region of operations for 2.7T (and possibly calculated regions for other magnitude of B)?
2. In discussion of using spin squeezing, was $\xi^2=4$ dB achievable experimentally? Reference suggested a squeezing of 0.5 dB. How much improvement would one obtain for 0.5 dB?
3. In the conclusion and discussion section, the author mentioned quantum sensing applications where large averaging times are limiting. However, the use of large measurement numbers in this enhanced protocol would sound contradictory to solving this issue.
4. In supplementary information there are tiny typos which I presume the authors will fix (see highlighted copy).

Reviewer #3

(Remarks to the Author)

In this manuscript, R. Maier et al. report a significant advance in solid-state spin metrology, providing the first experimental demonstration of projection-noise-limited readout in an NV ensemble. By successfully extending nuclear assisted repetitive measurement protocols previously established with single-spin systems, they achieve a 3.8 dB noise reduction below photon shot noise and explicitly probe the crossover into the quantum projection noise regime. Importantly, the direct access to these fundamental fluctuations opens new frontiers for solid-state spin based quantum sensing, potentially improving sensitivity by orders of magnitude. Overall, the experimental results are robust and compelling, and I would support the publication in Nature Communications if authors could address a few points detailed below.

1. While ^{14}N nuclei are used as effective two-level registers in this work, please comment whether this is an intentional choice over ^{15}N , and whether protocol and scaling would be the same.
2. An important practical consideration is the reliance on a 2.7 T bias field, which is not broadly accessible. Please estimate the minimum field required to (i) observe the crossover into the projection-noise limited regime and (ii) obtain a meaningful sensitivity gain.
3. Given the central role of state-selective MW and RF pulses and repeated mapping between electron and nuclear sublevels, please include a concise energy-level diagram showing relevant manifolds and the addressed transitions with frequencies (MW and RF). This would substantially improve readability and reduce ambiguity.
4. The manuscript states that the projection-noise-limited readout is “generally applicable to arbitrary ensemble sizes”. In practice, reaching the crossover requires a large collected photon number before relaxation (e.g. 2.7 T field used in this work). Given that the crossover already appears less evident for $N_{\text{NV}} \sim 170$ in supplementary data, please comment on the practical requirements for observing crossover with increasing ensemble size.

Minor points:

5. The notation alternates between up/down arrow and +1, 0, -1, for the same nuclear spin degree of freedom, which at times can be confusing. Please clarify and standardize notations.
6. In the Fig 4b caption, please clarify whether “the electron spin $\langle J_z \rangle$ ” stands for the nuclear variable as defined previously or an “effective polarization” standing in for the electron after mapping.
7. In the Supplementary Information, the sapphire substrate thickness has the wrong units, and multiple citations are not

properly displayed.

Version 1:

Reviewer comments:

Reviewer #1

(Remarks to the Author)

The rebuttal has addressed all of my questions from the first round of review. I appreciate the additional data presented showing the crosstalk of the ODMR pulses is minimised.

I recommend publication in Nature Communications of the updated manuscript. To reiterate, the methodology and conclusions are sound and there is sufficient detail to understand and potentially repeat the experiment in future.

Reviewer #2

(Remarks to the Author)

The author addressed my questions adequately. I recommend publication in Nat. Comm.

Reviewer #3

(Remarks to the Author)

The authors have satisfactorily addressed all my comments from the previous round. I have no further concerns and recommend acceptance.

The comments of the reviewers are reproduced in black, responses of the authors and, if applicable, changes are given in blue.

Response: We sincerely thank the reviewers for their thorough and insightful comments, which have improved the clarity and completeness of our manuscript. In response to the comments, we have revised the manuscript accordingly, which we believe addresses the raised concerns. Changes have been made to improve the consistency and clarity of our manuscript, for example by adding an energy level diagram to the supplementary information, including a new section about the magnetic field dependence of our protocol and streamlining the nomenclature of the spin levels throughout the manuscript. Further, we added additional information to the supplementary information regarding the different choices of nitrogen isotopes. Additionally, we added a data availability statement and author contributions.

Below we provide a point by point response to all questions raised:

Comments of Reviewer #1

Reviewer: The authors present work improving the sensitivity of readout of an ensemble of nitrogen vacancy nuclear spins, leveraging techniques employed for the readout of single spins. Operating at a higher than typical magnetic field, the photon shot noise is measured to be 3.8 dB lower than the spin-projection noise following repeated readout. This is significant as it opens up the possibility of observing spin squeezing in a room-temperature ensemble as well as measuring noise correlations across the ensemble. The latter is exemplified through a simple T2 relaxometry experiment examining the spin projection under a phase-randomised stimulus.

Overall, the work is reasonably presented and to my knowledge is the first demonstration of measurements of a room temperature spin ensemble at the quantum limit in the solid state. The choice of data collection and analysis methods are sound. I believe this manuscript is appropriate for this journal and will be useful to the community, though some minor revision may be needed.

Response: We are thankful to the referee for the thorough analysis of our manuscript and the positive evaluation.

Reviewer: Below are some technical questions and comments I have.

Reviewer: 1. For the ODMR detection of the nuclear spin states, a series of electron rotations were performed at 73 GHz. It wasn't mentioned what the Rabi frequency of the electron spin was, but the frequency upconversion and amplification chain, along with the pulse modulation can lead to spurious excitation of the neighbouring ESR lines (especially for a narrow separation of Δ = 2.16 MHz). Can the authors quantify this readout mapping error?

Response: We thank the reviewer for this comment, as cross talk in our MW pulses would lead to an artificial decrease of the observed spin projection noise, leading to wrong estimation of the number of observed NVs. Thus, we are using Rabi frequencies of about 0.4 MHz (duration of π pulses: $1.3 \mu\text{s}$) for our spin selective MW pulses, to allow a clear spectral separation of the individual transitions. Based on the Fourier limit of our rectangular pulses, we expect no significant crosstalk for a splitting of 2.16 MHz (see Fig. 1 added to this note). Additionally, we performed a measurement assessing the effect of n microwave π pulses applied at frequency

Figure 1: **Experimental evidence to exclude crosstalk of microwave drives.** a) Experimental ODMR spectrum including the theoretical frequency spectrum of a MW pulse centered at f_0 as a sinc function. b) Experimental data to exclude crosstalk when reading out neighbouring transitions. Since the readout contrast remains unaffected for $\Delta f = -2.16$ MHz even after $n = 100$ pulses, crosstalk is negligible. On resonance ($\Delta f = 0$), the spin state is fully decohered after $n = 100$ pulses. The measurement sequence is provided in the upper panel.

Δf from the left ODMR transition f_0 on that electron spin state, before reading it out selectively. If n pulses at frequency detuning Δf would drive the spin state, we would see an difference in the optical readout contrast compared to the case without applied pulses, as can be seen for the resonant case (zero detuning, $\Delta f = 0$). Even for $n = 100$ consecutive pulses, no effect for $|\Delta f| > 1$ MHz is visible. Thus, we conclude that spurious excitation does not play a significant role in our experiment and estimate the error per gate by this effect to less than 1%. We added information about the pulse duration and Rabi frequency to Supplementary Note 1.

Reviewer: On the topic of the readout, the perpendicular component of the hyperfine interaction was not listed, but the "T1" decay measurements are in fact dominated by hyperfine-enhanced relaxation, as opposed to direct spin-lattice relaxation. Were any measurements performed to measure the hyperfine strength transverse to the external field? This has significant implications in the discussion of the increasing number of readout shots performed, because this relaxation term scales with the number of shots m , rather than with time.

Response: We appreciate the reviewer bringing this ambiguity to our attention. The perpendicular hyperfine component A_{\perp} in ^{14}NV centers is typically given by $A_{\perp} = -2.6$ MHz (see e.g. Chen *et al.*, PRB 92, 020101 (2015)). In our manuscript, when we are mentioning T_1 of the nitrogen spin, we are not referring to the natural spin lattice relaxation, which can reach minutes at elevated fields. Instead, as correctly noted, our repetitive readout leads to a faster decay of the nitrogen polarization based on the hyperfine interaction, although the effect is minimized by application of a strong magnetic field well aligned to the principal axis of the NV center (see Neumann *et al.*, Science 329, 542 (2010)). As a result, we see an effective decay parametrized as a function of the performed readouts m (see Supplementary Fig. 7). This decay is what we refer to as effective T_1 in our work, even though it is, strictly speaking, not directly related to the natural spin lattice relaxation. To clarify this ambiguity, we added an explanation to Supplementary Note 4.

Reviewer: Finally, the T1 decay is shown to reduce the spin-projection noise due to depolar-

isation. Conceivably performing a negative result measurement, i.e. monitoring the expected dark-state, would reduce the impact of readout allowing the spin-photon level to be reduced further while maintaining the spin-polarisation. There is recent work on this performed on donor spins in silicon (see below).

Refs:

Avshalom C Elitzur and Lev Vaidman. Quantum mechanical interaction-free measurements. *Foundations of physics*, 23(7):987–997, 1993.

Maximizing the nondemolition nature of a quantum measurement via an adaptive readout protocol <https://arxiv.org/abs/2511.10978>

Response: The referee asked if a negative-measurement concept can help to overcome the effective T_1 depolarization due to the readout process. In itself this concept allows to minimize unwanted excitations of the system to an uncontrolled state and Hamiltonian, and was also used by our recent experiments with SiC at low temperature (see Hesselmeier *et al.*, PRL 132, 180804), and followed work on T-centers in silicon, using a negative “dark” state readout for a non perturbative “initialization” readout. However, in the current experiment, we operate in a room temperature setting, where the readout ability stems not from a selective optical excitation but rather inter system crossing induced spin dependent optical contrast, of around 30%. As such, readout of the spin state always involves optical excitation of the system, with the “less bright” state undergoing more non-radiative decay via a metastable state. This is in the stark difference to low temperature experiments with quantum dots / donors, or color centers/atoms/ions, where spin selective optical excitations or current tunneling with 100% contrast are possible. The T_1 process of the ancillary nuclear spin of NV center is known to be happening in the optically excited state, where the transverse hyperfine components increase from 2 to 50 MHz, which cannot be avoided in our case. In the end, the merit of our work is the ability to demonstrate a QND nature of the readout despite necessity to utilized disturbing “bright”, and “less bright” readout mechanism.

Reviewer: Detailed comments: Fig 2d, the green photon shot-noise curve should not have datapoints as it is not data.

Response: The photon shot noise depicted in 2 d) is actually directly measured by analyzing the width of the Skellam distribution obtained by subtracting the two reference readouts r_1 and r_2 and dividing it by the contrast $2knc$, as shown in Supplementary Fig. 5. We added an explanation to Supplementary Note 3.

Reviewer: The introduction of c , r_1 and r_2 is unclear. I believe r_1 and r_2 are the reference measurements bookending each sequence, but this needs to be explicit. The given form for c is not intuitive. Fig 3a, where are the two histograms derived from? It appears they are from different τ_{RF} slices, but this is not mentioned.

Response: We thank the reviewer for pointing this out. We adjusted the description of the parameters r_1, r_2 and c in the main text and added annotations to the histograms in Fig. 3a.

Comments of Reviewer #2

Reviewer: Several works in atomic physics have demonstrated the capability to beat the measurement shot noise and go beyond the standard quantum limit or the projection noise limit. However, solid-state systems, where readouts rely on fluorescence differences due to spin-dependent intersystem crossing rates, have been plagued by “low information” per photon count and thus more susceptible to photon shot noise. This work demonstrated one path towards reaching the spin projection noise limit by exploiting long-coherence-time nitrogen-14 nuclear spins and their hyperfine interactions with the electronic spins.

The authors worked with an NV center ensemble subjected to high magnetic field (~ 2.7 T). Nuclear-assisted readout protocol was used to measure the polarization of nitrogen-14 by mapping the N-14 spin population to the electron. While this is a conventional protocol for exploiting the weakly interacting N-14 as the long-lived “memory”, the authors explored the large field regime where they can afford a large number of readout cycles to average down the photon shot noise and reveals the cross-over from shot-noise-limited to spin-projection-noise-limited statistics. Having this tool at hand, the authors demonstrated a measurement of spatially correlated spin ensembles and an enhanced quantum tomography.

Overall, the paper is well-written and the derivations are sound. I would like to recommend the paper for publication in Nature Communications.

Response: We thank the reviewer for this positive evaluation and appreciate the feedback.

Reviewer: There are a few small questions that may improve the paper further:

1. Nuclear T_1 is extended significantly due to high magnetic field for repeated readouts. Would it be possible to show in Fig. 5 a) the region of operations for 2.7T (and possibly calculated regions for other magnitude of B)?

Response: This is an important point. In Fig 2 of this document, we overlaid the calculated quadratic increase of the nitrogen spin T_1 with Fig. 5 of the main text. It can be seen, that already for $B_0 < 0.5$ T the effective T_1 is long enough to allow for the $m > 3000$ readouts necessary, to achieve a possible sensitivity gain. We believe by adding the requested information, Fig. 5 would be too crowded, but added a remark about the necessary minimal magnetic field to the main text. Additionally, we added a new section to the Supplementary Information, discussing the dependency of the observed transition from photon shot noise to spin projection noise on the applied magnetic field (Supplementary note 6 in the updated manuscript).

Reviewer: 2. In discussion of using spin squeezing, was $\eta^2 = 4dB$ achievable experimentally? Reference suggested a squeezing of 0.5 dB. How much improvement would one obtain for 0.5 dB?

Response: To the best of our knowledge, $\eta^2 = 0.5$ dB is the largest experimental demonstration of spin squeezing in solid state systems, which suffer from a large degree of anisotropy in the hyperfine interaction (Wu *et al.*, Nature 646, 68 (2025)). However, in atomic systems up to 20 dB of spin squeezing have been demonstrated experimentally, e.g. Hosten *et al.*, Nature 529, 505 – 508 (2016). As such, the addition of the larger spin squeezing values in our discussion are seen as a potential outlook for the field. We added the requested sensitivity improvement for 0.5 dB squeezing to the main text at the end of the results section.

Figure 2: **Operational regimes for different B-fields.** A second axis indicating the T_1 of the nitrogen spin (in number of readouts m , x-axis) with respect to the applied B-field (y-axis) has been introduced.

Reviewer: 3. In the conclusion and discussion section, the author mentioned quantum sensing applications where large averaging times are limiting. However, the use of large measurement numbers in this enhanced protocol would sound contradictory to solving this issue.

Response: It is true, that at first glance the long readout times necessary for our protocol sound counterintuitive for facilitating fast measurements of physical signals. However, the gain in noise reduction overcompensates this by reducing the required number of averages. Here, the relevant figure of merit is the sensitivity η , which includes the necessary protocol time, as well as the noise reduction and provides information about how long a measurement has to be averaged (T_{meas}) to resolve a given signal strength δB with a signal-to-noise ratio of 1 ($\eta = \delta B \sqrt{T_{\text{meas}}}$). As such, an improved (lower) sensitivity strictly implies a reduced total measurement time to resolve a given signal. Since we calculate that our protocol can improve the sensitivity in certain regimes (Fig. 5), the total measurement times are also reduced there.

Reviewer: 4. In supplementary information there are tiny typos which I presume the authors will fix (see highlighted copy).

Response: We fixed the typos.

Comments of Reviewer #3

Reviewer: In this manuscript, R. Maier et al. report a significant advance in solid-state spin metrology, providing the first experimental demonstration of projection-noise-limited readout in an NV ensemble. By successfully extending nuclear assisted repetitive measurement protocols previously established with single-spin systems, they achieve a 3.8 dB noise reduction below photon shot noise and explicitly probe the crossover into the quantum projection noise regime. Importantly, the direct access to these fundamental fluctuations opens new frontiers for solid-state spin based quantum sensing, potentially improving sensitivity by orders of magnitude. Overall, the experimental results are robust and compelling, and I would support the publication in Nature Communications if authors could address a few points detailed below.

Response: We are grateful to the reviewer for their detailed analysis and positive feedback.

Reviewer: 1. While ^{14}N nuclei are used as effective two-level registers in this work, please comment whether this is an intentional choice over ^{15}N , and whether protocol and scaling would be the same.

Response: We thank the reviewer for bringing this lack of explanation to our attention. Indeed, as our protocol only utilizes two levels of the nitrogen spin, ^{15}N would be the logical choice. However, when the diamond sample was prepared, only ^{14}N was available, but the improvements we expect from using ^{15}N are only minor. One advantage would be the typically increased polarization of the nuclear spin when applying the cascaded polarization transfer scheme used in this work, which is faster and more robust when polarizing a two-level system compared to a three-level system, thus increasing the contrast in our later experiments that rely on it. In the earlier experiments about the thermal spin projection noise, only a minor influence is expected. As we are switching from a binned three-level $I = 1$ to a two-level $I = 0.5$ system, the observed spin projection noise remains almost constant, as

$$\sigma_{\tilde{j}_z}(^{14}\text{N}) \propto \sqrt{\frac{I(I+1)}{3}} \sqrt{\frac{1}{3}} \sim 0.47 \quad (1)$$

$$\sigma_{\tilde{j}_z}(^{15}\text{N}) \propto \sqrt{\frac{I(I+1)}{3}} = 0.5 \quad (2)$$

(see Supplementary Eq. 17). To clarify this point, we added an explanation to Supplementary Note 2.

Reviewer: 2. An important practical consideration is the reliance on a 2.7 T bias field, which is not broadly accessible. Please estimate the minimum field required to (i) observe the crossover into the projection-noise limited regime and (ii) obtain a meaningful sensitivity gain.

Response: Please see our answer to question 1 of reviewer #2.

Reviewer: Given the central role of state-selective MW and RF pulses and repeated mapping between electron and nuclear sublevels, please include a concise energy-level diagram showing relevant manifolds and the addressed transitions with frequencies (MW and RF). This would substantially improve readability and reduce ambiguity.

Response: We agree with the reviewer and added an energy level diagram and a description of the spin selective transitions to Supplementary note 1.

Reviewer: The manuscript states that the projection-noise-limited readout is "generally applicable to arbitrary ensemble sizes". In practice, reaching the crossover requires a large collected photon number before relaxation (e.g. 2.7 T field used in this work). Given that the crossover already appears less evident for $\text{NNV} \sim 170$ in supplementary data, please comment on the practical requirements for observing crossover with increasing ensemble size.

Response: As the spin projection noise scales as $\sqrt{\frac{1}{N_{\text{NV}}}}$ and the photon shot noise scales as $\sqrt{\frac{1}{n_{\text{photons}}}} \propto \sqrt{\frac{1}{N_{\text{NV}}}}$, there is no fundamental reason why the requirements (in terms of number of readouts) to observe the crossover should increase for larger ensembles of NV centers, as is mentioned in the main text. A re-plotted version of Supplementary Figure 5 b is added to this document. There it can be seen, that the crossover point experimentally does not shift to higher number readouts, i.e. it is not harder to reach for larger ensembles. More important

Figure 3: **Comparison of crossover point.** The dashed line indicates the crossover point where the photon shot noise is equal to the spin projection noise.

for the location of the crossover point are the spin properties in the observed ensemble. Since our sample was not perfectly homogeneous, we selected a spot with larger optical contrast in Region 2, thus reaching the crossover point earlier.

Reviewer: Minor points:

5. The notation alternates between up/down arrow and +1, 0, -1, for the same nuclear spin degree of freedom, which at times can be confusing. Please clarify and standardize notations.

Response: We corrected the notation in the main text and Supplementary Information, now using $|+1, 0, -1\rangle$ throughout the manuscript.

Reviewer: 6. In the Fig 4b caption, please clarify whether "the electron spin $\langle J_z \rangle$ " stands for the nuclear variable as defined previously or an "effective polarization" standing in for the electron after mapping.

Response: Here, $\langle J_z \rangle$ still stands for the nuclear variable as before and we adjusted the caption of Fig. 4b accordingly.

Reviewer: 7. In the Supplementary Information, the sapphire substrate thickness has the wrong units, and multiple citations are not properly displayed.

Response: We fixed the mentioned points in the Supplementary Information.